# Glutacetine^®^ Biostimulant Applied on Wheat under Contrasting Field Conditions Improves Grain Number Leading to Better Yield, Upgrades N-Related Traits and Changes Grain Ionome

**DOI:** 10.3390/plants10030456

**Published:** 2021-02-28

**Authors:** Victor Maignan, Patrick Géliot, Jean-Christophe Avice

**Affiliations:** 1Normandie Univ, UNICAEN, INRAE, UMR EVA, SFR Normandie Végétal FED4277, Esplanade de la Paix, 14032 Caen, France; 2Via Végétale, 44430 Le Loroux-Bottereau, France; pgeliot@viavegetale.com

**Keywords:** plant biostimulants, nitrogen fertilizer, *Triticum aestivum*, nitrogen use efficiency, ionome, grain quality, SPAD-502^®^, X-ray fluorescence

## Abstract

Wheat is one of the most important cereals for human nutrition, but nitrogen (N) losses during its cultivation cause economic problems and environmental risks. In order to improve N use efficiency (NUE), biostimulants are increasingly used. The present study aimed to evaluate the effects of Glutacetine^®^, a biostimulant sprayed at 5 L ha^−1^ in combination with fertilizers (urea or urea ammonium nitrate (UAN)), on N-related traits, grain yield components, and the grain quality of winter bread wheat grown at three field sites in Normandy (France). Glutacetine^®^ improved grain yield via a significant increase in the grain number per spike and per m^2^, which also enhanced the thousand grain weight, especially with urea. The total N in grains and the NUE tended to increase in response to Glutacetine^®^, irrespective of the site or the form of N fertilizer. Depending on the site, spraying Glutacetine^®^ can also induce changes in the grain ionome (analyzed by X-ray fluorescence), with a reduction in P content observed (site 2 under urea nutrition) or an increase in Mn content (site 3 under UAN nutrition). These results provide a roadmap for utilizing Glutacetine^®^ biostimulant to enhance wheat production and flour quality in a temperate climate.

## 1. Introduction

Chemical fertilizers, especially nitrogen (N), are one of the main inputs for boosting yield, but also one of the most expensive inputs in terms of economic costs and environmental risks. Many crops require large amounts of this element to maximize yield [1,2], but the use of N fertilizers is implicated in damage to the environment and human health. The main environmental impacts of N are in the contamination of surface and groundwater resources and greenhouse gas emissions [3,4,5]. In terms of human health, the effects depend on the accumulation of nitrate in plant tissue; when nitrate is reduced to nitrite in the human body, it can cause methemoglobinemia, which is dangerous for young children [6]. Nitrite can also react with several chemical compounds (amines and amides), producing N-nitrous compounds, suspected as probable carcinogens in humans [7].

For these reasons, it is necessary to choose the appropriate chemical form and the correct dose and application time to better manage N fertilizer application. These practices contribute to improvements in N use efficiency (NUE), which is linked to the capacity of plants to take up, transport, store and recycle N [8,9,10,11]. NUE is in general expressed as the harvestable yield per N supply [12]. Approaches for improving NUE are mainly based on biotechnology and plant breeding, as well as chemical molecules such as N-(n-butyl) thiophosphoric triamide (NPBT) and 3,4-dimethypyrazole phosphate (DMPP) [13,14,15,16], which are urease and nitrification inhibitors, respectively. However, these inhibitors may affect the activities of soil microorganisms [15,17] and also have negative impacts on plant metabolism [18,19] and human health. This is why it is necessary to find alternatives that are environmentally friendly, such as treatment with plant biostimulants [20]. 

According to the EU regulation 2019/1009, “a plant biostimulant is a product that stimulates plant nutrition processes independently of the product’s nutrient content with the sole aim of improving one or more of the following characteristics of the plant or the plant rhizosphere: (a) nutrient use efficiency; (b) tolerance to abiotic stress; (c) quality traits; and (d) availability of confined nutrients in the soil or rhizosphere”. Plant biostimulants are defined more by the plant response they elicit than by their composition [21]. They can influence phenotypic traits and improve yield by enhancing crop stress tolerance and nutrient uptake and assimilation. Therefore, bioactive substances (humic acids, protein hydrolysates, amino acids, seaweed extracts, etc.) and microorganisms (mycorrhizal fungi, plant growth-promoting rhizobacteria (PGPR), etc.) are gaining ground in agricultural systems because of their potential to improve nutrient use efficiency. Previous studies have documented that the application of plant biostimulants improves leaf pigmentation and photosynthetic efficiency and triggers several molecular and physiological processes that induce an improvement in NUE, yield and quality [22,23,24,25]. The capacity of biostimulants to improve NUE is the main reason for their adoption across the agricultural sector, considering their economic and environmental advantages [26]. These products can therefore be used to complement fertilizers in order to reduce inputs and increase NUE [27] in crops such as winter wheat.

Wheat (*Triticum aestivum* L.) is one of the most important cereal crops. Winter wheat cultivars are commonly grown in France due to their good adaptation to the oceanic temperate climate. The grain yield values of 7 t ha^−1^ [28] are among the highest in the world and France is the fifth largest producer globally [29]. It is well documented that biostimulants are able to improve wheat productivity and quality. For example, the application of fulvic acid extracts improved soil properties and enhanced the growth, yield and nutritional status of wheat grown on contrasting soil types in China [30]. In the same region, the application of different kinds of biostimulants alongside zinc (Zn) increased Zn concentrations and bioavailability in wheat grain [31]. Research from India [32] that aimed to increase micronutrient concentrations in wheat grains so that human dietary requirements can be met has reported increased wheat yields and biofortification following interactions between bacteria from native soil and arbuscular mycorrhizal fungi. In Iran, fluorescent *Pseudomonas* has been shown to enhance phosphorus (P) uptake (+42%) and grain yield (+17%) under water-deficit stress [33]. In the European context, recent field studies in Poland have demonstrated the effect of silicon (Si) on the growth of spring wheat [34], and additions of protein hydrolysate have improved yield and grain quality of winter wheat [35]. Closer to French weather conditions, *Bacillus* strains applied to winter wheat in Belgium improved wheat performance under different N supplies [36], and in France itself, marine and fungal biostimulants have led to increased grain yields (+1.8 to +5.5%) in durum wheat [37]. However, there have been few studies conducted in France and other temperate climate regions on the effects of biostimulant application on bread wheat productivity. Moreover, the performance of such treatments is dependent not only on the date and the dose of the application, but also on the soil and weather conditions [38]. To the best of our knowledge, there have not been any studies assessing the impact of an organo-mineral complex biostimulant, such as Glutacetine^®^, on the NUE, grain yield and quality of bread wheat under field conditions in temperate regions.

Regarding quality traits, plant biostimulants induce changes in wheat grain technological characteristics [35], the grain proteome [39] and especially the grain ionome [32]. Aside from the presence of sufficient calories and proteins in the diet, another important trait is the micronutrient composition of food. Indeed, malnourishment may arise from micronutrient deficiency. According to the World Health Organization (2002), iron (Fe) and Zn are deficient in the diet of more than half of the world’s population [40,41]. Further, magnesium (Mg), and calcium (Ca) are also deficient in the diet of some of the population [42,43]. Although manganese (Mn) deficiency in humans is not a widespread global issue, this trace element also plays pivotal roles in growth and development as a cofactor or a component of several key enzyme systems [44]. These micronutrient deficiencies have been amplified by a constant decline in mineral concentrations in grain over the last century, and have impacted the Fe, Zn [41] and Mg composition at harvest [43]. It has been postulated that this loss of harvest quality has been the result of varietal selection over the past 70 years that aimed to achieve higher yield under a sufficient macronutrient supply and despite sufficient micronutrient content in the soil [45,46]. Fe, Zn, Ca, Mn and Mg are also of great importance for plant metabolism, especially for enzymes and regulatory proteins [47], and are also essential to signaling [48,49], transport [50,51,52] and remobilization [53,54]. This means that both harvest quality and crop development can be limited by their lack of availability. Another problem that aggravates this situation is the presence of phytate in wheat grains, an antinutritional factor that forms complexes with minerals such as Ca, Mg, Zn and Fe, thus reducing their bioavailability in flour [55,56].

To prevent micronutrient deficiencies, wheat biofortification programs have been developed mainly for Zn [57,58,59] and Fe [41,60], but also for Mn [61]. Research in this domain tends to focus on either breeding selection or fertilizing practices. Research in plant breeding with the aim of increasing uptake and translocation towards the grain can take many years before new varieties are obtained for food production. Another strategy relies on the development of enhanced-efficiency fertilizers and biostimulants that are more efficient than traditional chemicals. Designing and developing new biostimulants or enhanced fertilizers is a crucial process that requires accurate testing of the product’s effects on crop yield and quality traits and a deep understanding of their mechanism of action under field conditions.

The aim of this study was to assess the effect of foliar application of Glutacetine^®^ on winter wheat grown under field conditions with different N fertilizer regimes (urea alone or urea–ammonium–nitrate solution (UAN)). Experiments were carried out in three different sites in Normandy (France) to evaluate the possible beneficial effects of this biostimulant on three different cultivars grown under optimal N conditions to determine its influence on NUE, yield and grain quality, as well as the grain and straw ionomes.

## 2. Results

### 2.1. Effects of Site, N Fertilizer Forms and Glutacetine^®^ on Grain Yield and Its Components

To better understand the impact of site (S), N fertilizer forms (F), the Glutacetine^®^ treatment (T) and their interactions, a global analysis of variance (ANOVA) was first undertaken for grain yield and several yield components (spike number per square meter, grain number per spike, grain number per square meter, thousand grain weight (TGW) and specific weight). The global ANOVA presented in Table 1 shows a strong effect of field site on grain yield and yield components (*p* < 0.01), which was probably due to the contrasting pedoclimatic conditions and the different cultivars used (Appendix A). On the other hand, there was no effect of N fertilizer forms or S × F × T interactions on grain yield and its components (Table 1). Considering each of the N fertilizer forms (urea or UAN), the effect of site on grain yield and its components was confirmed by ANOVA, except for grain number per spike, which was not significant under urea nutrition (Table 1). Nevertheless, we showed a fertilizer effect on the specific weight at site 1 (*p* < 0.05) due to a decrease in this parameter when wheat was fertilized with UAN (irrespective of the biostimulant treatment) compared to urea nutrition (Figure 1F, *p* = 0.0139).

Regarding biostimulant application (T), the global ANOVA indicated a significant effect of Glutacetine^®^ on grain number per spike (+7.3%, *p* = 0.045, Table 1) and grain number per square meter (+3.3%, *p* = 0.033, Table 1), and a slight effect on other yield components: spike number per square meter (*p* = 0.066) and TGW (*p* = 0.055). Interestingly, with urea nutrition, we showed a significant treatment effect on spike number per square meter, grain number per spike and TGW (*p* < 0.05, Table 1), while it was significant for grain number per square meter under UAN fertilizer (*p* = 0.030, Table 1). Indeed, Glutacetine^®^ decreased spike number per square meter and increased grain number per spike in all situations except under UAN nutrition at site 3 (Figure 1B and C), leading to a higher grain number per square meter (except under urea fertilizer at site 3), which was slightly significant under UAN nutrition at site 2 (*p* = 0.082, +9.4%, Figure 1D). This parameter was significantly increased by Glutacetine^®^ at this site, irrespective of the N fertilizer forms (+7.7%, *p* = 0.039, Figure 1D). In the same way, TGW was slightly improved by Glutacetine^®^ treatment at all sites (Figure 1E and Table 2). It was significantly higher at site 3 under urea nutrition (+5.5%, *p* < 0.05) and irrespective of the type of N fertilizer (*p* < 0.05, Figure 1E). However, Glutacetine^®^ application did not induce consistent change in specific weight (Figure 1F). The amelioration of grain number per square meter (+751 g m^−^²) and TGW (+2 g) after Glutacetine^®^ application led to an improvement in grain yield (*p* = 0.076, Table 2) by 345 kg ha^−1^ under urea nutrition and by 387 kg ha^−1^ under UAN nutrition, while grain yield ranged from 10.17 to 13.22 t ha^−1^ over the three sites of experimentation (Figure 1A).

### 2.2. Impacts of Site, N Fertilizer Forms and Glutacetine^®^ on N-Related Traits

After the final harvest, global ANOVA revealed strong site and fertilizer effects (*p* < 0.001 for both) for N concentration in the grain (Table 1). The site effect was observed under both urea and UAN nutrition (Table 1), while the fertilizer effect was significant only at sites 1 and 3 (Figure 2A). Indeed, the N content in grain ranged from 1.70 (site 1) to 2.25% (site 2) and site 3 had intermediate values (Figure 2A). The fertilizer effect highlighted a better N content in the grain when urea was used compared to UAN at sites 1 (+10.6%, *p* = 0.005) and 3 (+13.1%, *p* = 0.015). Focusing on the effect of Glutacetine^®^, we did not observe consistent change in N concentration in the grain. Furthermore, the analysis of N content in straw also indicated an important site effect irrespective of the fertilizer used, while neither fertilizer nor biostimulant had an impact on this parameter (Table 1). In fact, site 3 presented much lower straw N concentrations than sites 1 and 2 (Figure 2B).

Moreover, the total grain N was determined to evaluate the amount of N exported to grains depending on the different field conditions. Regardless of the type of N fertilizer used (urea or UAN), ANOVA still indicated a site effect on total grain N (*p* < 0.001, Table 1). Compared to sites 1 and 3, more N was exported to grains of wheat at site 2 (+64 kgN ha^−1^, *p* < 0.001, Figure 2C). There was also an effect from the type of fertilizer (*p* < 0.01), especially at site 1, highlighting the better efficiency of urea compared to UAN in exporting N to grains (Figure 2C). Even though global ANOVA did not reveal a significant effect of Glutacetine^®^ on total grain N (*p* = 0.166), there was a strong tendency for this parameter to improve. Indeed, compared to control, total grain N was slightly increased by Glutacetine^®^ (to 9.6%: +18.5 kgN ha^-1^ at site 3 under UAN nutrition), irrespective of the site and the N form (Figure 2C). These results were then supported by a calculation of NUE that also showed an important site effect regardless of the type of N fertilizer (*p* < 0.001, Table 1). NUE was significantly different at each site (*p* < 0.001 between them, Figure 2D): site 1 had the highest NUE, site 3 an intermediate NUE and site 2 the lowest, confirming the contrasting pedoclimatic, cultivar and N management conditions. As for total grain N, three-way ANOVA indicated that Glutacetine^®^ slightly improved NUE (*p* = 0.077, Table 1 and Figure 2D). This means that for each kilogram of N applied, spraying Glutacetine^®^ facilitated the production of an additional 2 kg of grain. In comparison to the control groups, it could be hypothesized that this improvement in total grain N was mainly due to a more efficient N uptake and/or utilization (i.e., assimilation and/or remobilization) after spraying Glutacetine^®^.

To address this question, the N status in leaves was studied in response to the biostimulant. In order to determine the leaf N status, measurement of the chlorophyll level (the SPAD index) was used as a means to estimate relative N content in the flag leaf. For that, the %N in the flag leaf was first determined by IRMS from a calibration curve at site 1. Relative N content in leaves was then correlated with the SPAD index and the correlations were assessed with Pearson’s test. We showed a strong relationship between these two parameters (R² = 0.77, *p* < 0.0001, Figure 3A), which allowed us to calculate the relative N content in the flag leaf at the heading stage (BBCH51) and during seed development (BBCH71, Figure 3B and C). Irrespective of the fertilizer and the biostimulant treatment, three-way ANOVA (Table 1) indicated a site effect on the %N in the flag leaf at both stages. Indeed, at the heading stage, the flag leaf had a higher N content at site 2 compared to sites 1 and 3 (Figure 3B), suggesting that this flag leaf presents a higher activity in terms of photosynthesis. During seed development (Figure 3C), the %N in the flag leaf was also different depending on the site, but the ANOVA of urea nutrition (ANOVA Urea) did not show consistent change, whereas the ANOVA of UAN nutrition (ANOVA UAN) revealed a slight site effect (*p* = 0.076, Table 1). We then calculated the difference in variation in %N (∆%N, Figure 3D) between the heading and seed development stages to assess whether plants continued to accumulate N in the flag leaf or whether they had commenced the remobilization process. Three-way ANOVA indicated a significant site effect (*p* < 0.001) on the ∆%N in the flag leaf irrespective of the fertilizer (Table 1). Indeed, ∆%N at site 2 resulted in negative values (meaning that N remobilization had started), while the values at sites 1 and 3 were positive (Figure 3D). Three-way ANOVA also revealed an S × T interaction (*p* < 0.05, Table 1), which was mainly due to the biostimulant effect at site 3 under both urea and UAN nutrition (*p* < 0.01, Figure 3D). Remarkably, with urea nutrition, there was a slight biostimulant effect (i) at site 2 (*p* = 0.081), which decreased more the ∆%N in the flag leaf compared to the control (−0.25% with Glutacetine^®^ versus +0.05 for control, Figure 3D), and (ii) at site 3 (*p* = 0.058), which increased more than the ∆%N in the flag leaf compared to the control (0.27 with Glutacetine^®^ instead of 0.12, Figure 3D), and maintained higher levels of chlorophyll and N in these leaves.

### 2.3. Effects of Site, N fertilizer Forms and Glutacetine^®^ on the Grain Ionome

For the study of the grain ionome, we then measured elemental concentrations in wheat grain using X-ray fluorescence (XRF) calibrated with HR ICP-MS (Appendix A). As a rapid and non-destructive method of elemental analysis, XRF allowed the detection and quantification of Ca, Fe, K, Mg, Mn, P, S and Zn in grain. A site effect was observed for Ca (*p* < 0.001), Fe (*p* < 0.06), K (*p* < 0.001), S (*p* < 0.001) and Zn (*p* < 0.01) contents in wheat grain (three-way ANOVA, Table 2). Considering N fertilizer forms, the site effect was also significant for Ca, K, P, S and Zn under urea supply and for Ca and S under UAN fertilizer (two-way ANOVA, Table 2). Indeed, the Ca and S contents in the grain were the highest at site 3 and the lowest at site 1 (Table 3). Furthermore, we observed a fertilizer effect only for P and S contents (three-way ANOVA, Table 2). In fact, the fertilizer effect on P concentration was mainly observed at site 1 (UAN fertilizer reduced P content by 9.1% compared to urea, Table 3). This is why there were S × F interactions for P and S concentrations (*p* = 0.079 and *p* < 0.05, respectively, Table 2). A similar S × F interaction was observed for K content (*p* = 0.018), which was especially affected by the N form at site 1 (Table 2): urea treatment led to a higher K concentration in the grain compared to UAN (+12.7%, Table 3).

Focusing on Glutacetine^®^ effects, three-way ANOVA revealed an S × F × T effect for Fe (*p* = 0.077), Mn (*p* < 0.01) and S (*p* < 0.05, Table 2). Considering UAN fertilizer, ANOVA UAN indicated a slight S x T effect on Fe (*p* = 0.073, Table 2) which was mainly due to the reduction in Fe content in the grain by Glutacetine^®^ under UAN fertilizer at site 3 (−8.3 ppm, *p* = 0.027, Table 3). Regarding Mn content, there was an S × T interaction revealed in ANOVA Urea (*p* = 0.058), which was due to the significant increase in Mn content by 5.8 ppm after spraying Glutacetine^®^ with UAN at site 3 (Table 3). In addition, under urea nutrition, Glutacetine^®^ reduced P content at site 2 and we observed the same effect for Mg content at site 1 under UAN fertilizer (Table 3). These results demonstrated that the Glutacetine^®^ biostimulant may change micronutrient concentrations in winter wheat grain under field conditions.

### 2.4. Effects of Site, N fertilizer Forms and Glutacetine^®^ on the Straw Ionome

Alongside analysis of the grain, we used XRF for determination of Ca, Fe, K, Mg and S concentrations in wheat straw after calibration with HR ICP-MS (Appendix A). Global ANOVA and ANOVA Urea showed a strong site effect for Ca, Fe, K and S (*p* < 0.05 minimum), while ANOVA UAN indicated a site effect for Ca and K (*p* < 0.001 for both) with a slight impact on S content (*p* = 0.058, Table 2). Indeed, the Ca and S contents ranged from their lowest values at site 1 to the highest at site 3 (Table 4). In the same way, the K concentration in straw was substantially different depending on the site (from 5700 ppm at site 3 to 15,712 ppm at site 2, Table 4). In contrast, the Fe concentration was higher at site 1 (157 ppm on average) compared to sites 2 (121 ppm on average) and 3 (116 ppm on average, *p* = 0.014, Table 4). However, there was no consistent change regarding the effect of N fertilizer form.

Remarkably, there was a slight Glutacetine^®^ effect (*p* = 0.053) and a significant S × T interaction (*p* < 0.05) for the Ca content in straw (Table 2). This observed S × T interaction (Table 2) was mainly due to the biostimulant effect at site 3, which significantly reduced the Ca concentration by −9.7% irrespective of the fertilizer (two-way ANOVA, *p* = 0.034, Table 4). ANOVA UAN also showed an S × T interaction for Ca (Table 2) because Glutacetine^®^ significantly reduced Ca content at site 3 under UAN nutrition (−9.7%, *p* = 0.020, Table 4). Regarding urea fertilizer, a significant Glutacetine^®^ effect on Fe content in straw was observed (two-way ANOVA, *p* = 0.012, Table 2), which tended to decrease after the application of the formulation (Table 4). Altogether, these results suggested that Glutacetine^®^ initiated changes in the straw ionome, especially in terms of Ca and Fe content, and that this was dependent on the site and N management, which could have impacts on remobilization efficiency.

## 3. Discussion

The objectives of this study were to provide details about the effects of Glutacetine^®^ on grain yield components, N-related traits and bread wheat grain quality under contrasting field conditions. Clearly, the present work highlighted the impacts of this biostimulant in combination with urea or UAN fertilizer applications on grain yield components, total grain N, NUE and especially the grain ionome.

### 3.1. Glutacetine^®^ Improves Grain Yield by Increasing Grain Number per Spike and per Square Meter

The present work was performed in three contrasting agro-pedoclimatic conditions and strong site effects on grain yield and its components were evident, regardless of the form of N nutrition (Table 1 and Figure 1). These different impacts were mainly related to soil type, cultivar and weather conditions (Appendix A).

According to the literature, nutrient accumulation and yield of winter wheat are impacted by N supply [62] and N fertilizer form [63]. In the present study, we confirmed that the N form influenced TGW (site 3) and specific weight (site 1) but did not affect grain yield in winter wheat (Table 1). As UAN is composed of 50% urea and the N supply was the same at each site, the different forms of fertilizer management were not sufficient to significantly change yield.

Considering the three field experiments together, Glutacetine^®^ had a significant positive effect on grain number expressed per spike and per square meter (especially at site 2, Figure 1D), irrespective of the form of N fertilizer (Table 1). This is in accordance with the increase in grain yield components obtained under controlled conditions by Maignan et al. (2020), who also used spray application on winter wheat, but UAN was supplied as a foliar application. In our field trials, the N supply provided was close to wheat demand, while in the controlled conditions experiment, a lower dose was provided in the system. However, this grain yield augmentation is quite similar to the increases obtained with protein hydrolysate under field conditions in Poland [35], but higher than those obtained after foliar application of fulvic acid, seaweed extract, amino acids, microbial incubates [31] or Si [34]. PGPR might also have significant results in wheat, but mainly under abiotic stresses like water deficiency [33] or in the absence of N fertilizer [36]. Very interestingly, in controlled conditions, we have also reported that sprayed Glutacetine^®^ significantly increased grain number per spike (Maignan et al., 2020). The present study confirmed the same level of amelioration under contrasting field conditions, and Glutacetine^®^ seemed more efficient with urea (+9.6%) than UAN nutrition (+5.1% compared to the control). This could be due to the better fruiting efficiency [64,65] with Glutacetine^®^, which might have a strong impact on spike fertility [39]. In accordance with our findings, PGPR may also increase the number of grains per spike in wheat [32]. Nevertheless, there was no consistent change in the number of grains per ear following the application of protein hydrolysate [35]. The positive effect of this kind of biostimulant was also observed in crops such as rice, finger millet, radish and cowpea where soil application was more efficient than foliar application [66]. 

Focusing on other yield components, Glutacetine^®^ improved TGW under urea nutrition and especially at site 3 (Table 1 and Figure 1E), which is characterized by a superficial clay limestone soil type. This result seems surprising in light of our recent study performed on winter wheat under controlled conditions (Maignan et al., 2020), which demonstrated the negative effect of spraying Glutacetine^®^ on TGW. This difference in the effect of Glutacetine^®^ between the two experiments may be explained by the significant impact of weather conditions and soil type in the present work (Appendix A) versus the well-watered conditions of the greenhouse experiments. In addition, other biostimulants such as fulvic acid and PGPR have also enhanced TGW in sandy loam and clay loam soils in winter wheat [30,32]. A mixture of bioactive compounds was also able to increase TGW in maize [67]. Thus, the present work indicated that spraying Glutacetine^®^ has similar effects to other biostimulants, leading to an improvement in seed yield components.

### 3.2. Glutacetine^®^ Affects NUE and Total Grain N Differently

Because organic matter was applied at site 1, N fertilizer was higher at site 2 and the previous crop at site 3 was a leguminous crop (*Medico sativa*) (Appendix A), it was evident that these contrasting conditions strongly influence N-related traits in winter wheat. Indeed, we observed significant site effects on the %N in the flag leaf (Figure 3B and C), the grain and the straw, and this induced different levels of total grain N and NUE, irrespective of the N fertilizer form (Figure 2). However, the form of N fertilizer only had a strong effect on the N content in grain (*p* < 0.001) and on the total grain N (*p* < 0.01, Table 2). The high yield potential at site 2 due to its soil characteristics (see details in Appendix A) and the high N inputs (see details in Appendix A) might explain the lower effect of the N type at this site (Figure 2 and Figure 3). However, the N concentration in the grain was reduced when wheat was fertilized with UAN compared to urea at each site (Figure 2A). Thus, this study demonstrated that using urea fertilizer on wheat is more efficient than UAN.

In our field experiments, it seemed that the impact of Glutacetine^®^ on NUE was dependent on the site characteristics. Indeed, in comparison to the control, the increase in NUE (Figure 2D) by Glutacetine^®^ observed at site 2 with urea nutrition was associated with a reduction in the %N in straw (Figure 2B) and a significant decrease in the %N in the flag leaf, as observed between heading and seed development (Figure 3D). These data suggest that the increase in the NUE by Glutacetine^®^ (+7.2%) may be due to an improvement in the remobilization of N from leaves and straw. On the other hand, the present work also showed that Glutacetine^®^ maintained high N content in the flag leaf at site 3, irrespective of the form of N fertilizer (Figure 3D). This outcome could be due to maintenance of post heading N uptake as demonstrated by Maignan et al. (2020) in winter wheat cultivated under controlled conditions. This result also suggests that the Glutacetine^®^ supply increases the leaf life span via the maintenance of photosynthesis and/or delaying the foliar senescence. This last hypothesis is in agreement with previous studies showing that biostimulant applications such as Si that can delay leaf senescence in several plant species like oilseed rape [68,69], strawberries [70] and wheat [34]. PGPR might also enhance chlorophyll levels in leaves of wheat plants [32,33]. Moreover, we have shown that Glutacetine^®^ slightly enhanced the total grain N and NUE in winter wheat, irrespective of the N form (Table 1). As previously reported [68,71], biostimulants can promote the expression of genes encoding for N transport and assimilation, which lead to an improvement in N uptake, especially after the heading stage [39]. Better N remobilization efficiency is also important for increasing total grain N and NUE [72]; several biostimulants such as marine and fungal biostimulants improved total grain N [37], while others like PGPR are able to increase NUE [73] in wheat. Altogether, these results suggest that Glutacetine^®^ applied on urea- or UAN-fertilized wheat crops enhances total grain N and NUE, probably due to an effect on N uptake and N remobilization efficiency after the heading stage. 

### 3.3. Glutacetine^®^ Induces Changes in the Grain and Straw Ionomes

Mineral composition represents a major criterion of quality in bread wheat grain. While we did not show consistent change regarding ash content (data not shown), which is an indicator of mineral content that is important for human nutrition [74], the development of rapid and non-destructive XRF analyses of the grain ionome helped us to demonstrate the site dependency of the Ca, K, S and Zn contents in the wheat grain (Table 2). These site effects were more significant when urea was used compared to UAN, including for P (Table 2). Regarding the straw ionome, Ca, K, S and Fe content were also dependent on the site. These differences were mostly due to soil characteristics (Appendix A). Indeed, in our field conditions, Ca availability in the soil varied by a factor of 6 (2.1‰ at site 1, 3.6‰ at site 2 and 12.3‰ at site 3), which is strongly correlated with the Ca content in the grain and straw (Table 3 and Table 4). However, while exchangeable K was lower at site 1 compared to the other two sites, the wheat grain and straw from this site did not have the lowest K content. There are two explanations for this finding: either the different wheat cultivars used at each site have contrasting K use efficiency (uptake and remobilization) and/or the organic matter input at site 1 released large amounts of K during the crop cycle, which compensated for the low levels of exchangeable K in the soil. In addition, Zn content in the grain was lower at site 2, probably because of the high yields, which induced a dilution of this microelement. Moreover, it is well documented that pH influences the bioavailability of chemical elements such as Fe in the soil and plays an important role in their availability to plants [75]. The soil concentration and availability of mineral elements for plants generally decrease as the pH and Ca content increase [76,77]. In the present work, sites had contrasting pH (6.1, 7.3 and 8.4 at sites 1, 2 and 3, respectively) and CaO content (Appendix A), which could explain the higher Fe content in the straw from site 1.

Considering the multi-elemental composition of Glutacetine^®^ (Table 5), it can be suggested that its function is to enhance nutrient uptake from the environment, as well as assimilation and translocation to grains, as this study showed (Table 3 and Table 4). This is in agreement with published data about the action of other biostimulants in increasing plant mineral uptake and improving nutrient use efficiency [26,78,79]. The decrease in P content is in agreement with our previous work, which showed this effect with another Glutacetine^®^-derived formulation [39] and may be associated with lower phytate content with antinutritional properties [80]. Such variations in mineral composition and grain content are also a function of the biostimulant used and the crop tested. For example, seaweed extract has been reported to enhance the P concentration in winter wheat, but no consistent change was observed for K and Zn, while fulvic acid has been associated with an increase in Fe content [31]. PGPR was able to increase P, Ca, Zn, Cu, Fe and B content in wheat grains [32] whereas protein hydrolysate only affected Cu [35]. In other species, a mixture of bioactive compounds induced a higher N, P and K content in maize [67], while Si supply improved the Fe concentration in grains of winter oilseed rape under two contrasting N fertilizer treatments [69]. On the other hand, an improvement in remobilization of these nutrients due to Glutacetine^®^ spraying cannot be excluded (especially for Ca from straw at site 3 with UAN nutrition, Table 5), allowing the element concentration to be maintained in grain despite increases in yield. However, we could expect a higher response of wheat plants in terms of Ca content in grain because this element is important in the Glutacetine^®^ formulation (15.6%DW, Table 5). The increase in nutrient concentrations in seeds might also be a consequence of stimulated root uptake after the heading stage, as demonstrated previously for N [39].

These results are particularly interesting for Mn because this micronutrient is important for plant metabolism and also for human nutrition and health [61,81,82]. Indeed, Mn has crucial roles as a cofactor or component of several key enzyme systems [44]. In our work, at site 3 under UAN nutrition, the application of Glutacetine^®^ increased grain yield by 724 kg ha^−1^ (Figure 1A), decreasing Fe content but not Zn content, whereas it significantly improved the Mn content (Table 3). Thus, to confirm this biofortification effect, further investigations will be necessary to verify whether Glutacetine^®^ can stimulate Mn transporters in source organs during senescence, as observed for other biostimulants [50,51], and/or induce transcription factors such as no apical meristem (NAM) proteins that regulate the transport of this micronutrient to the grains [53].

## 4. Materials and Methods

### 4.1. Field Experiment Design

During autumn 2018, three different cultivars of winter wheat (*Triticum aestivum* L., cv. Sacremento, Libravo and Chevignon) were sown at three different sites in Normandy, France (see Appendix A for more details), which belongs to the oceanic temperate region. Plot size was 24 m^2^ (12 m × 2 m) and three replicates (plots) were used per treatment on each site. The adopted model of the field experiment was the randomized complete block design with three main blocks. From April 2019–June 2019, systematic applications of plant protection treatments were carried out. Weather conditions (average temperature and total rainfall) were from 11.9 to 12.4 °C and from 588 to 611 mm between September 2018 and August 2019 (Appendix A). Previous crops and soil properties were very different at each site (Appendix A).

All plots were fertilized with 30 kg S ha^−1^ at the tillering stage (BBCH 29, [83]). N fertilizer applications were intended to replicate farmer practice and were calculated using the method (the so-called “Méthode du Bilan”) described by the French COMIFER committee [84]. This method measures the amount of mineral N in the soil in February to estimate the total soil N supply. With the estimation of total plant needs, the method allows the application of a N dose that is close to crop demand (Appendix A). Two different N fertilizers were used: 46% urea and a 39% ammonium nitrate urea solution (UAN). The N fertilizer dose was applied at 152, 210 and 152 kg N ha^−1^ at sites 1, 2 and 3, respectively (Appendix A). The first fertilizer dose was applied at tillering (BBCH 21), the second at stem elongation (BBCH 31) and the rest at the flag leaf stage (BBCH 39, Appendix A). For the last N application, we followed the new N requirement indicators (“bq” coefficient) for growing each bread wheat cultivar with the dual objective of optimum yield and grain protein content in line with market requirements [85]. For the biostimulant treatment, a Glutacetine^®^ formulation (Table 5) was sprayed on each fertilizer type on the same day as the last N application (BBCH 39) to maximize the N uptake in plants. Then, 5 L ha^−1^ of Glutacetine^®^ were mixed with water for spraying. The levels of chlorophyll in the flag leaf were determined via the SPAD index measured at heading and during seed development using a SPAD-502^®^ chlorophyll meter (Konica Minolta Inc., Tokyo, Japan). Ear number was counted in each plot of the three sites and grain and straw were collected using a combine harvester. All samples were dried (60 °C) for 48h and then ground to a fine powder for elemental analysis.

### 4.2. Seed Yield Components and N Use Efficiency

After harvesting of winter wheat, TGW (g), grain number per spike, grains per square meter, specific weight (grain density, kg hL^−1^) and moisture (%) were determined. Yields were then calculated with standard moisture (15%) and NUE according to Moll et al. (1982) [12] using the following equation: NUE (kg kg^−1^) = Grain yield (kg ha^−1^)/N supply (kg ha^−1^).

### 4.3. Elemental Analysis

For N analysis in leaves, grain and straw, aliquots of each dried sample were placed into tin capsules using a microbalance and the total N concentration was determined with a continuous flow isotope ratio mass spectrometer (IRMS, Horizon, NU Instruments, Wrexham, United Kingdom) linked to a C/N/S analyzer (EA3000, Euro Vector, Milan, Italy). Total grain N was then calculated (kgN ha^−1^).

The grain and straw of winter wheat were also subjected to multi-elemental analysis (macronutrients and micronutrients) using the XRF technique (X-ray fluorescence). Because whole grain reference samples were not available, it was necessary to develop a set of calibration and validation standards for this study. A collection of bread wheat samples with variation in element concentrations was selected from a previous wheat experiment [39]. Selected samples were analyzed by a high-resolution inductively coupled plasma mass spectrometry (HR ICP-MS, Thermo Scientific, Element 2^TM^, Bremen, Germany). Dry matter of each of the samples (40 mg) was suspended with 800 µL of concentrated HNO_3_, 200 µL of H_2_O_2_ and 1 mL of Milli-Q water. All samples were then spiked with two internal standard solutions containing gallium, rhodium and iridium with final concentrations of 10 and 2 µg L^−1^, respectively. After microwave acidic digestion (Multiwave ECO, Anton Paar, les Ulis, France), all samples were diluted with 50 mL of Milli-Q water to obtain solutions containing 2% (*v*/*v*) nitric acid. Before HR ICP-MS analysis, samples were filtered at 0.45 µm using a Teflon filtration system (Digifilter, SCP Science, Courtaboeuf, France). Quantification of each element was performed using external standard calibration curves and concentrations were expressed in ppm. For each element, linear regression and correlation between XRF and HR ICP-MS data were assessed by Pearson’s test using XLSTAT software (Addinsoft, France). Significant correlations were obtained for K, P, Mg, S, Ca, Fe, Zn and Mn in grain samples (Appendix A). Except for Zn, all of these elements and Ni correlated in straw samples (Appendix A). Subsequently, all grain and straw samples were placed in sample cups and element concentrations were determined with an XRF analyzer (Portable XRF S1 TITAN 800, Bruker, Kalkar, Germany). Quantification of each element was performed using calibration curves of samples quantified by HR ICP-MS (Appendix A).

### 4.4. Statistical Analysis

All statistical analyses were performed with R Statistical Software version 3.3. The normality of data and the homogeneity of variance were verified using the Shapiro and Levene tests, respectively. When data followed a normal distribution, one-, two- or three-way analysis of variance (ANOVA) was used to evaluate the effect of site (S), fertilizer forms (F) (i.e., urea or UAN solution), Glutacetine^®^ (treatment, T) and their interactions (S × F, S × T, F × T and S × F × T, Appendix A). The resulting variations in data are expressed as the mean ± standard error (SE) for *n* = 3 and significantly different means between treatments were separated with Fisher’s test (*p* ≤ 0.05). Some sets of data were transformed using the Box–Cox method in order to obtain a normal distribution and apply ANOVA and Fisher’s test. When variables did not satisfy normality due to culture effects, the non-parametric Kruskal–Wallis test was used (*p* ≤ 0.05).

## 5. Conclusions

This study has established that the spraying of Glutacetine^®^ onto winter wheat under field conditions leads to significant amelioration of grain yield components, leading to better yield. In our field conditions, the improvement in N-related traits (mainly total grain N and NUE) is probably due to a better post-heading N uptake and/or a better N remobilization efficiency, as observed in the leaf N status. In addition, Glutacetine^®^ might promote Mn biofortification in the winter wheat grain. Moreover, XRF is a promising tool for rapid analysis of the grain and straw ionomes and could have value as an efficient diagnostic tool to support sustainable agriculture. Indeed, knowledge of the effectiveness of mineral fertilizers as well as the action of biostimulants in improving fertilizer efficiency is essential to achieve significant increases in yield and the quality of harvested products. Altogether, the results confirmed the importance of Glutacetine^®^ application for farmers who use urea or UAN fertilizer on winter wheat crops, and the use of the SPAD-502^®^ chlorophyll meter to follow N-related traits and XRF to analyze wheat grain and straw ionomes. 

## Figures and Tables

**Figure 1 plants-10-00456-f001:**
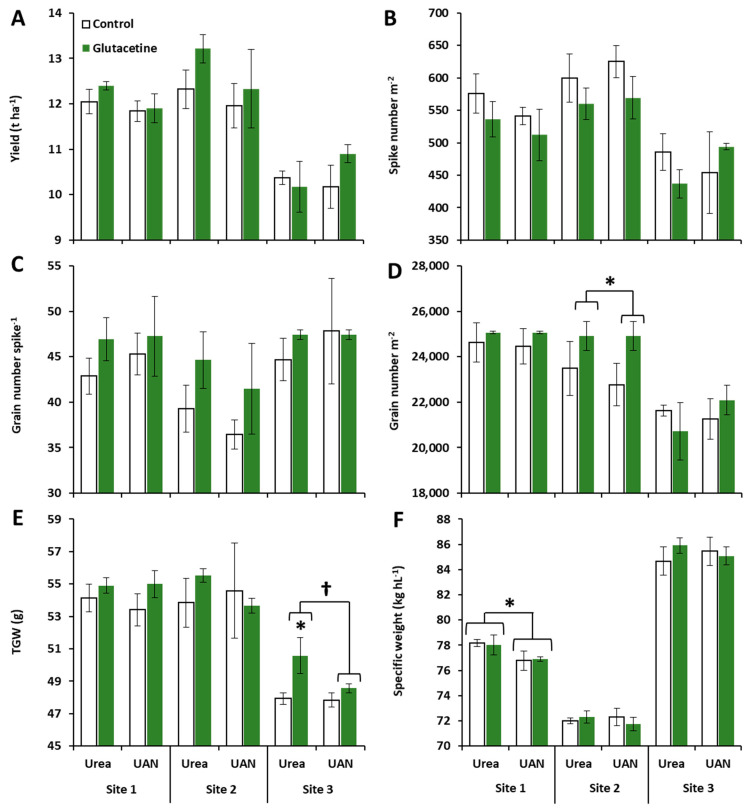
Effect of the Glutacetine^®^ formulation under urea or UAN nutrition on grain yield and yield parameters in wheat (*Triticum aestivum* L.). (**A**) Grain yield (t ha^−1^), (**B**) spike number per square meter, (**C**) grain number per spike, (**D**) grain number per square meter, (**E**) thousand grain weight (TGW) and (**F**) specific weight (kg hL^−1^). Plant culture was carried out under three different field conditions in France, Normandy. N was provided at tillering, stem elongation and the flag leaf stages. Glutacetine^®^ was mixed with water and sprayed at a dose of 5 L ha^−1^ on the same day as the last N application. Bars indicate means ± SE. ⁎ and † denote significant differences according to Fisher’s test or Kruskal–Wallis test, respectively (*p* < 0.05; *n* = 3).

**Figure 2 plants-10-00456-f002:**
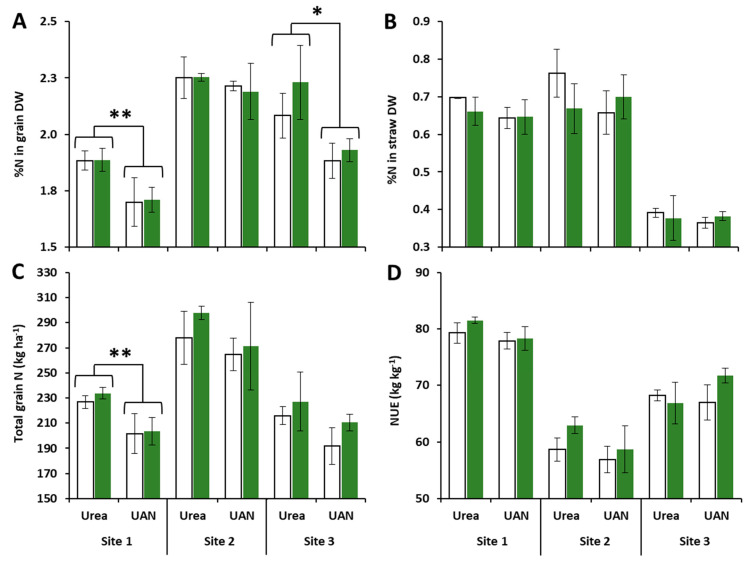
Effect of Glutacetine^®^ formulation under urea or UAN nutrition on N-related traits at maturity in wheat (*Triticum aestivum* L.). (**A**) %N in grain dry weight (DW), (**B**) %N in straw DW, (**C**) total grain N (kg ha^−1^) and (**D**) N use efficiency (NUE, kg kg^−1^). Plant culture was carried out under three different field conditions in France, Normandy (otherwise, see Figure 1 for more details). Bars indicate means ± SE. ⁎ denotes significant differences according to Fisher’s test (⁎: *p* < 0.05; ⁎⁎: *p* < 0.01; *n* = 3).

**Figure 3 plants-10-00456-f003:**
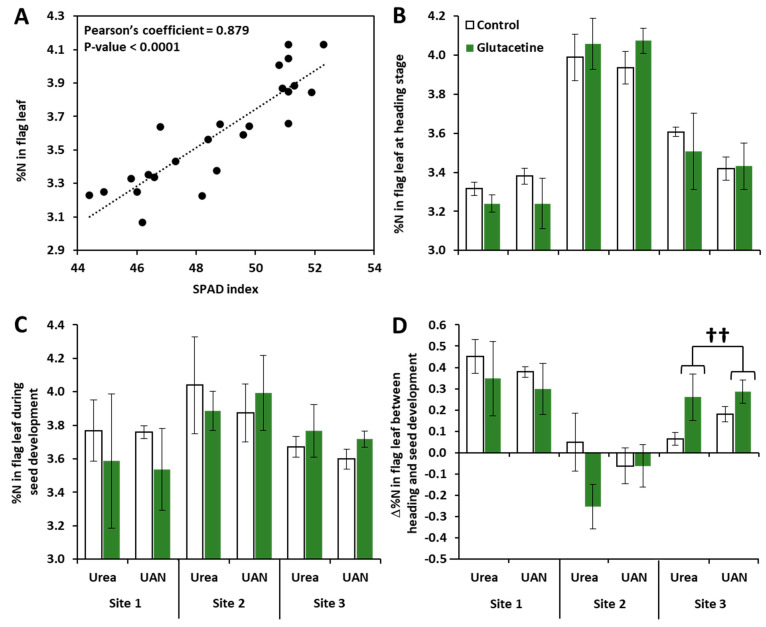
SPAD index calibration and N-related traits associated with the flag leaf in wheat (*Triticum aestivum* L.). (**A**) Relationship between the SPAD index (chlorophyll level) and %N in the flag leaf of wheat, (**B**) %N in the flag leaf at heading stage, (**C**) %N in the flag leaf during seed development and (**D**) the Δ%N in the flag leaf between heading and seed development stages of wheat. Plant culture was carried out under three different field conditions in France, Normandy (otherwise, see Figure 1 for more details). Bars indicate means ± SE. †† denotes significant differences according to the non-parametric Kruskal–Wallis test (*p* < 0.01; *n* = 9 for the ∆%N in the flag leaf between heading and seed development stages).

**Table 1 plants-10-00456-t001:** Effects of site (S), form of N nutrition (F: urea or urea–ammonium–nitrate (UAN)) and/or biostimulant treatment (T: Glutacetine^®^) on grain yield, yield components and N-related traits determined by three-way analysis of variance (global ANOVA) and two-way ANOVA (for each form of N fertilizer).

		Grain Yield and Its Components							N-Related Traits						
		Yield (t/ha)	Spikes/m²	Grains/Spike	Grains/m²	TGW	Specific Weight		%N in Grain	%N in Straw	Total Grain N	NUE	%N in the Flag Leaf at Heading	%N in the Flag Leaf during Seed Development	Δ%N in the Flag Leaf
**Global ANOVA**	**Site (S)**	⁎⁎⁎	⁎⁎⁎	⁎⁎	⁎⁎⁎	⁎⁎⁎	⁎⁎⁎		⁎⁎⁎	⁎⁎⁎	⁎⁎⁎	⁎⁎⁎	⁎⁎⁎	⁎	⁎⁎⁎
	**Fertilizer (F)**								⁎⁎⁎	⁎⁎					
	**Treatment (T)**	<0.08	<0.07	⁎	⁎	<0.06						<0.08			
	**S × F**														
	**S × T**														⁎
	**F × T**														
	**S × F × T**														
**ANOVA Urea**	**Site**	⁎⁎⁎	⁎⁎⁎		⁎⁎⁎	⁎⁎⁎	⁎⁎⁎		⁎⁎⁎	⁎⁎⁎	⁎⁎⁎	⁎⁎⁎	⁎⁎⁎		⁎⁎⁎
	**Treatment**		⁎	⁎		⁎									
	**S × T**														<0.08
**ANOVA UAN**	**Site**	⁎⁎	⁎⁎	⁎	⁎⁎⁎	⁎⁎⁎	⁎⁎⁎		⁎⁎⁎	⁎⁎⁎	⁎⁎⁎	⁎⁎⁎	⁎⁎⁎	<0.08	⁎⁎⁎
	**Treatment**				⁎										
	**S × T**														

⁎, ⁎⁎, ⁎⁎⁎ = significant at 0.05, 0.01 and 0.001, respectively, *p* > F for *n* = 3. ∆%N in the flag leaf corresponds to the variation of %N in the flag leaf between heading and seed development (*n* = 9). NUE: nitrogen use efficiency; TGW: thousand grain weight.

**Table 2 plants-10-00456-t002:** Effects of site (S), form of N fertilizer (F: urea or urea–ammonium–nitrate (UAN)) and/or biostimulant treatment (T: Glutacetine^®^) on the grain and straw ionomes determined by three-way analysis of variance (global ANOVA) and two-way ANOVA (for each form of N fertilizer).

		Grain Ionome									Straw Ionome				
		Ca	Fe	K	Mg	Mn	P	S	Zn		Ca	Fe	K	Mg	S
**Global ANOVA**	**Site (S)**	⁎⁎⁎	<0.06	⁎⁎⁎				⁎⁎⁎	⁎⁎		⁎⁎⁎	⁎	⁎⁎⁎		⁎⁎
	**Fertilizer (F)**						⁎	⁎							
	**Treatment (T)**										<0.06				
	**S × F**			⁎			<0.08	⁎							
	**S × T**										⁎				
	**F × T**														
	**S × F × T**		<0.08			⁎⁎		⁎							
**ANOVA Urea**	**Site**	⁎⁎⁎		⁎⁎⁎			⁎⁎	⁎⁎⁎	⁎		⁎⁎⁎	⁎⁎⁎	⁎⁎⁎		⁎
	**Treatment**											⁎	<0.07		
	**S × T**					<0.06									
**ANOVA UAN**	**Site**	⁎⁎⁎						⁎⁎⁎	<0.06		⁎⁎⁎		⁎⁎⁎		<0.06
	**Treatment**														
	**S × T**		<0.08					<0.06			⁎				

⁎, ⁎⁎, ⁎⁎⁎ = significant at 0.05, 0.01 and 0.001, respectively, *p* > F for *n* = 3.

**Table 3 plants-10-00456-t003:** Effects of Glutacetine^®^ on the grain ionome under contrasting field (sites 1, 2 and 3) and N fertilizer conditions (with urea or urea–ammonium–nitrate (UAN)). The relative content (expressed as ppm) of each element was determined by X-ray fluorescence (XRF) and then calculated with inductively coupled plasma mass spectrometry (ICP-MS) calibration.

			P	K	S	Mg	Ca	Fe	Zn	Mn
**Site 1**	**Urea**	**Control**	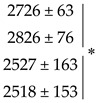	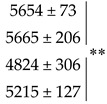	1112 ± 10	1154 ± 45	616 ± 19.2	34.0 ± 5.6	37.7 ± 1.7	30.6 ± 2.8
		**Glutacetine^®^**			1105 ± 29	1278 ± 99	608 ± 10.1	26.8 ± 8.1	36.7 ± 1.4	38.7 ± 3.7
	**UAN**	**Control**			1083 ± 8	1273 ± 61	546 ± 24.0	29.3 ± 4.4	35.3 ± 4.2	34.9 ± 2.5
		**Glutacetine^®^**			1080 ± 37	1187 ± 12 **†**	608 ± 36.1	30.9 ± 1.4	38.6 ± 3.0	33.3 ± 1.8
**Site 2**	**Urea**	**Control**	2667 ± 18	4957 ± 114	1274 ± 19	1227 ± 76	677 ± 20.9	33.8 ± 2.9	34.8 ± 3.8	30.8 ± 1.6
		**Glutacetine^®^**	2541 ± 34 ⁎	4899 ± 59	1240 ± 16	1118 ± 58	677 ± 14.5	30.8 ± 1.9	33.8 ± 2.7	31.7 ± 1.6
	**UAN**	**Control**	2659 ± 60	5337 ± 381	1259 ± 7	1264 ± 82	708 ± 22.1	32.2 ± 3.1	34.8 ± 1.7	31.3 ± 3.6
		**Glutacetine^®^**	2621 ± 143	5054 ± 313	1264 ± 2	1186 ± 29	681 ± 27.1	34.2 ± 2.5	32.8 ± 4.4	31.5 ± 2.1
**Site 3**	**Urea**	**Control**	2532 ± 100	5971 ± 326	1285 ± 30	1147 ± 49	829 ± 41.5	34.8 ± 3.9	40.8 ± 3.1	35.7 ± 3.5
		**Glutacetine^®^**	2595 ± 24	5858 ± 167	1316 ± 24	1192 ± 58	803 ± 37.8	37.3 ± 0.2	40.8 ± 3.4	32.0 ± 2.0
	**UAN**	**Control**	2369 ± 62	5650 ± 143	1267 ± 13	1216 ± 124	792 ± 9.4	39.7 ± 2.3	42.3 ± 2.2	30.8 ± 1.6
		**Glutacetine^®^**	2560 ± 184	5603 ± 465	1239 ± 10	1287 ± 83	778 ± 51.4	31.4 ± 1.9 ⁎	39.1 ± 2.1	36.6 ± 1.3 ⁎

Values indicate means ± SE. ⁎ and **†** denote significant differences according to Fisher’s test or Kruskal–Wallis test, respectively (⁎, **†**: *p* < 0.05; ⁎⁎: *p* < 0.01; *n* = 3).

**Table 4 plants-10-00456-t004:** Effects of Glutacetine^®^ on the straw ionome under contrasting field (sites 1, 2 and 3) and N fertilizer conditions (with urea or urea–ammonium–nitrate (UAN)). The relative content (expressed as ppm) of each element was determined by XRF and then calculated with ICP-MS calibration.

			K	Mg	S	Ca	Fe
**Site 1**	**Urea**	**Control**	10945 ± 632	937 ± 58.9	866 ± 24.3	2639 ± 53	152 ± 3.6
		**Glutacetine^®^**	11231 ± 315	895 ± 52.6	852 ± 33.3	2726 ± 106	135 ± 9.0
	**UAN**	**Control**	11008 ± 461	826 ± 113.1	897 ± 72.4	2664 ± 86	144 ± 18.1
		**Glutacetine^®^**	10062 ± 406	835 ± 123.6	918 ± 27.4	2638 ± 34	196 ± 21.9
**Site 2**	**Urea**	**Control**	14796 ± 480	829 ± 48.6	964 ± 25.5	3487 ± 125	111 ± 14.9
		**Glutacetine^®^**	15630 ± 1155	711 ± 141.9	1004 ± 41.0	3236 ± 201	78 ± 6.0
	**UAN**	**Control**	16132 ± 1990	714 ± 39.5	949 ± 72.0	3269 ± 189	132 ± 55.8
		**Glutacetine^®^**	16289 ± 1405	764 ± 76.7	971 ± 35.3	3394 ± 93	162 ± 71.1
**Site 3**	**Urea**	**Control**	4692 ± 785	686 ± 149.6	1008 ± 75.9	4845 ± 320	131 ± 14.7
		**Glutacetine^®^**	6946 ± 1168	858 ± 82.2	977 ± 102.0	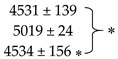	119 ± 8.9
	**UAN**	**Control**	5862 ± 965	975 ± 111.3	1050 ± 93.7		116 ± 9.2
		**Glutacetine^®^**	5299 ± 431	791 ± 135.2	1034 ± 38.1		98 ± 1.5

Values indicate means ± SE. ⁎ denotes significant difference according to Fisher’s test (*p* < 0.05; *n* = 3).

**Table 5 plants-10-00456-t005:** The chemical composition of Glutacetine^®^.

**Glutamic acid (%DW)**		3.6
**Organic acids (%DW)**		7.4
**Total Soluble Sugars (%DW)**		10
**Elements (%DW)**	**Cl**	19.8
	**Ca**	15.6
	**C**	6.26
	**N**	0.76
	**K**	0.31
	**Na**	0.31
	**Mo**	0.15
**Elements (mg kg^−1^ DW)**	**S**	300
	**B**	30
	**Mg**	19
	**Si**	14
	**P**	9.5
	**Cu**	2.6
	**Ni**	1.8
	**Co**	0.6
	**Zn**	0.3
	**Se**	0.08

## Data Availability

The data presented in this study are available on request from the corresponding author.

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
