# Peer review of "Glutacetine® Biostimulant Applied on Wheat under Contrasting Field Conditions Improves Grain Number Leading to Better Yield, Upgrades N-Related Traits and Changes Grain Ionome"

_plants, 2021, doi:10.3390/plants10030456_

Round 1

Reviewer 1 Report

Review of manuscript plants-1106136

The Authors have investigated an interesting topic – effect of Glutacetine Biostymulant application on wheat yield and the other plant characteristics. The objectives of this article are adequate and appropriate in view of the subject matter. Unfortunately, the manuscript contains serious methodological errors and should be significantly improved.

  • First of all, the results are only one-year. For me it is a methodological errors, even if the study concern 3 different sites.
  • The statistical methods are well described, but I have doubts whether it is necessary to present and interpret as many as three analyzes of variance (one, two and three-way ANOVA). The high number of statistical approaches adopted by the authors indicate the willingness to prove, by force, that the biostimulant has a significant effect. Perhaps, only 1 or 2 analysis (three-way ANOVA, one-way ANOVA) is / are enough to describe the obtained results.
  • The global ANOVA shows that there was no significant interaction of S x F x T. So why are lowercase letters used to denote homogeneous groups ? (Figures 1, 2, 3). In addition, post-hoc analysis was used, with a choice of Fischer's test. With Tukey’s procedure (HSD) it is more difficult to find a difference than with Fisher’s protected LSD. Perhaps, there can only be SE values.
  • If it was necessary to perform the non-parametric Kruskal-Wallis test, it should be indicated for which plant parameters it was performed and test results.
  • The manuscript can be shortened because the analysis of relationship between the HR ICP-MS and XRF methods is unnecessary, it was not the aim of the study. Such relationships may be a scientific problem for another publication.
  • The title should be changed. It suggests a positive significant effect of the Glutacetine Biostymulant, and in fact it has not been proven.
  • The description of figure 1 appears out of order as from Fig 1F
  • The units for the content of components in the Glutacetine Biostymulant should be standardized. Currently there are % and g / L. The table shows that the main compounds, or elements, in the biostymulant were Ca and Cl. Right ? If so, consider the function of these components in the discussion.
  • Instead of ppm, give mg / kg, and specify whether it is dry matter, or fresh matter
  • Line 602, editing error, flat = flag

Author Response

First of all, the results are only one-year. For me it is a methodological error, even if the study concern 3 different sites.

Unfortunately, we could not repeat the experiment for a second year. Nevertheless, as you mentioned, we had 3 different sites (and different cultivars) and we tested the effect of the biostimulant with two different fertilizer forms.

Perhaps, only 1 or 2 analysis (three-way ANOVA, one-way ANOVA) is / are enough to describe the obtained results.

As suggested, we have removed the two-way ANOVA for each experimental site as we did not observe many significant biostimulant effects. However, we decided to keep the two-way ANOVA for each form of N fertilizer because biostimulant effect was significant, especially for grain yield components.

The global ANOVA shows that there was no significant interaction of S x F x T. So why are lowercase letters used to denote homogeneous groups? (Figures 1, 2, 3).

We changed our statistic methodology to indicate only significant changes in Figures 1, 2 and 3.

In addition, post-hoc analysis was used, with a choice of Fischer's test. With Tukey’s procedure (HSD) it is more difficult to find a difference than with Fisher’s protected LSD. Perhaps, there can only be SE values.

We decided to use Fisher’s test because this test is especially recommended in studies with small sample sizes (https://doi.org/10.1073/pnas.1915454117): this was the case for our field experiments. In agreement with our statistical methodology, several recent studies published in Plants also used this statistical test. For instance, Fisher’s test was recently used in articles for study on wheat under field conditions (https://doi.org/10.3390/plants10010101), for the evaluation of strains (https://doi.org/10.3390/plants9121755) and for spinach under potassium deficiency (https://doi.org/10.3390/plants9121739). Others works evaluating the effects of fertilizer or biostimulant published in Plant Science (https://doi.org/10.1016/j.plantsci.2017.02.004) or Frontiers in Plant Science (https://doi.org/10.3389/fpls.2017.01265), including our recent publication (https://doi.org/10.3389/fpls.2020.607615), used Fisher’s test.

If it was necessary to perform the non-parametric Kruskal-Wallis test, it should be indicated for which plant parameters it was performed and test results.

We indicated when we used the non-parametric Kruskal-Wallis test for each figure and table.

The manuscript can be shortened because the analysis of relationship between the HR ICP-MS and XRF methods is unnecessary, it was not the aim of the study. Such relationships may be a scientific problem for another publication.

We agree with the comment that XRF developments are not the aim of the study. Nevertheless, we decided to keep the analysis of relationship between the HR ICP-MS and XRF methods because we think it is essential to explain our methodology. As suggested by the reviewer, we put the figures with the data showing the relationships between the HR ICP-MS and XRF methods in supplemental figures S2 and S3 in order to shorten the manuscript. We have also suppressed the reference about XRF data in the title (see comments just below).

The title should be changed. It suggests a positive significant effect of the Glutacetine Biostimulant, and in fact it has not been proven.

We changed the title: “Glutacetine® Biostimulant Applied on Wheat under Contrasting Field Conditions Improves the Grain Number leading to Better Yield, Upgrades N-related Traits, and Changes Grain Ionome”.

The description of figure 1 appears out of order as from Fig 1F

We changed the description of Figure 1 (L171-178).

The units for the content of components in the Glutacetine Biostimulant should be standardized. Currently there are % and g / L. Instead of ppm, give mg / kg, and specify whether it is dry matter, or fresh matter.

We standardized the units for the content of components in the Glutacetine Biostimulant (see revised Table 5, L573).

The table shows that the main compounds, or elements, in the biostimulant were Ca and Cl. Right? If so, consider the function of these components in the discussion.

We discussed this point, especially for Ca (see L514-516 in the revised manuscript)

Line 602, editing error, flat = flag

We corrected this editing error.

Reviewer 2 Report

Dear Authors,

The subject of the study is interesting and topical, with high scientific and practical importance.

The introduction is in accordance with the subject and correctly presented. Numerous scientific articles of recent date and in concordance to the topic of the study were consulted.

Methodology of the study was clearly presented, and appropriate to the proposed objectives.

The obtained results have been analyzed and interpreted in accordance with the current methodology.

The discussions are appropriate, in the context of the results, and was conducted compared to other studies in the field.

The scientific literature, to which the reporting was made, is recent and representative in the field.

However, the review of the article revealed some minor issues, which were noted in the article.

References

According to Instructions for Authors, Plants journal

"References should be described as follows, depending on the type of work:

Journal Articles:

  1. Author 1, A.B.; Author 2, C.D. Title of the article. Abbreviated Journal Name Year, Volume, page range.

Revision and correction is recommended.

  1. Makino, A. Photosynthesis, grain yield, and nitrogen utilization in rice and wheat. Plant Physiol. 2011, 155, 125–129, doi:10.1104/pp.110.165076.

Instead of

„1. Makino, A. Photosynthesis, Grain Yield, and Nitrogen Utilization in Rice and Wheat. PLANT PHYSIOLOGY 2011, 155, 125–129, doi:10.1104/pp.110.165076.”

Correct abbreviation of the name of some journals

e.g.

Page 25, Rows 739 - 740: "Plant Physiol." instead of “PLANT PHYSIOLOGY

More suggestions and corrections were made in the article.

According to Instructions for Authors, Plant journal, it is not recommended to use capital letters for each word in the title of the articles.

It is recommended to check and use the Plant journal template; Styles / MDPI_7.1_References

Author Response

References

According to Instructions for Authors, Plants journal

"References should be described as follows, depending on the type of work:

Journal Articles:

  1. Author 1, A.B.; Author 2, C.D. Title of the article. Abbreviated Journal Name Year, Volume, page range.

Revision and correction is recommended.

  1. Makino, A. Photosynthesis, grain yield, and nitrogen utilization in rice and wheat. Plant Physiol. 2011, 155, 125–129, doi:10.1104/pp.110.165076.

Instead of

„1. Makino, A. Photosynthesis, Grain Yield, and Nitrogen Utilization in Rice and Wheat. PLANT PHYSIOLOGY 2011, 155, 125–129, doi:10.1104/pp.110.165076.”

Correct abbreviation of the name of some journals

e.g.

Page 25, Rows 739 - 740: "Plant Physiol." instead of “PLANT PHYSIOLOGY

More suggestions and corrections were made in the article.

We corrected all these editing error in references.

According to Instructions for Authors, Plant journal, it is not recommended to use capital letters for each word in the title of the articles.

It is recommended to check and use the Plant journal template; Styles / MDPI_7.1_References

All the last articles in Plants journal use capital letters for each word in the title.

Reviewer 3 Report

The paper is well explained and could be useful for researchers who wish to evaluate biostimulant effects on grain yield components of wheat under contrasting field conditions. However, there are 2 questions and one suggestion I would like to make that I think could improve the document:

1- Upon reaching biostimulant application time, the fertilizer application can be replaced with a formulation entitled more suitable for plant development (due to the composition of Glutacetine®). Reasons?
2- Two phenological phases were evaluated: BBCH 21, BBCH39. what is your justification  (references) for these stages?

Biostimulants could directly influence the physiological activity in plant growth. If authors have measured physiological parameters such as net photosynthesis and stomatal conductance,  could improve the discussion part.

Author Response

1- Upon reaching biostimulant application time, the fertilizer application can be replaced with a formulation entitled more suitable for plant development (due to the composition of Glutacetine®). Reasons?

We have not really understood that is suggested by the reviewer 3 in this comment. So, we could not give a clear answer.

2- Two phenological phases were evaluated: BBCH 21, BBCH39. what is your justification (references) for these stages?

We evaluated the biostimulant effect at heading stage and during seed development because these two phenological phases correspond to the crucial stages for N uptake and translocation to the grain, as these stages are just before and after flowering.

Biostimulants could directly influence the physiological activity in plant growth. If authors have measured physiological parameters such as net photosynthesis and stomatal conductance, could improve the discussion part.

We agree that physiological parameters such as net photosynthesis and stomatal conductance are important. Nevertheless, in these three field trials we have not the facilities for doing these kinds of studies. On the other hand, we have used a SPAD device that is a rapid tool for the determination of chlorophyll level that is related to N level in flag leaf (see Fig. 3). Generally, a higher level of chlorophylls is also related to a higher photosynthesis activity and vice versa. This point was mentioned in the revised manuscript (see L245-248: “Indeed, at the heading stage, the flag leaf had a higher N content at site 2 compared to sites 1 and 3 (Figure 3B) suggesting that this flag leaf presents a higher activity in terms of photosynthesis.”)

Reviewer 4 Report

In the methodology of this research, it is necessary to emphasize the adopted model of the field experiment was the Randomized Complete Block Designs (RCB) not Completely Randomized Designs (CRD).

It may be worth emphasizing in the conclusions of the research that despite the described benefits of the Glutacetine® application, a significant increase in the yield of wheat grain was not achieved (it was only for site (S) but conditions there were significantly different).

Author Response

In the methodology of this research, it is necessary to emphasize the adopted model of the field experiment was the Randomized Complete Block Designs (RCB) not Completely Randomized Designs (CRD).

As recommended, we emphasized our adopted model as follows: “The adopted model of the field experiment was the Randomized Complete Block Design with three main blocks.” (see L. 539-541 of the revised manuscript).

It may be worth emphasizing in the conclusions of the research that despite the described benefits of the Glutacetine® application, a significant increase in the yield of wheat grain was not achieved (it was only for site (S) but conditions there were significantly different).

We revised our conclusions to describe only the significant benefits of Glutacetine®.

Reviewer 5 Report

The article should be simplified as it is too long and unclear.

Lines 131-136: repetition of the sentences written for the aim and in materials and methods.

Table 1 shall be placed in the materials and methods and not in the results.
  The results are not clear and the anova table (Table 2) is too commented on; on the contrary, the figures must be commented on.
The anova divided by fertilisers and sites does not seem to provide any particular information, especially for sites. Figures 1 2 and 3 represent third-order interactions with the letters of significance, but none of them is significant; the main effects should be reported when significant.
Also in Tables 4 and 5 the third order interaction is significant only for Fe, Mn and S (grain ionome).
The experimental conditions (e.g., Fig 1 lines 183-186) shall not be reported in the caption of the figures 1, 2 and 3.   There are also many repetitions in the discussions and in any case they must be rewritten because they report too many results.   The conclusions must also be rewritten because they report too many results

Author Response

The article should be simplified as it is too long and unclear.

Lines 131-136: repetition of the sentences written for the aim and in materials and methods.

We removed theses sentences.

Table 1 shall be placed in the materials and methods and not in the results.

We placed Table 1 in the materials and methods section.

The results are not clear and the anova table (Table 2) is too commented on; on the contrary, the figures must be commented on. The anova divided by fertilisers and sites does not seem to provide any particular information, especially for sites. 

We removed the two-way ANOVA for each experimental site as we did not observe many significant biostimulant effects. However, we decided to keep the two-way ANOVA for each form of N fertilizer because biostimulant effect was significant, especially on grain yield components.

Figures 1 2 and 3 represent third-order interactions with the letters of significance, but none of them is significant; the main effects should be reported when significant.

We changed our statistic methodology to indicate only significant changes in these figures.

Also in Tables 4 and 5 the third order interaction is significant only for Fe, Mn and S (grain ionome).

We changed our statistic methodology to indicate only significant changes in these tables.

The experimental conditions (e.g., Fig 1 lines 183-186) shall not be reported in the caption of the figures 1, 2 and 3.

We reported the experimental conditions only in Figure 1 and we shortened this part in Figures 2 and 3.

There are also many repetitions in the discussions and in any case, they must be rewritten because they report too many results. The conclusions must also be rewritten because they report too many results.

We revised our discussions and conclusions avoiding the repetitions in order to simplify and shorten the article.

Round 2

Reviewer 1 Report

Review of manuscript plants-1106136

The manuscript has been revised according to my comments and suggestions. However, there are still some errors in the manuscript:

Line 127, after (UAN) editorial error, two times parenthesis

Line 179, Line 335  under the tables there are a capital letter "P", elsewhere there is a lower case letter "p". Symbols needs to be unified.

Figure 1B still has the lowercase letters. They shouldn't be there because there was no interaction S x F x T (this is confirmed in Table 1). In my pdf version it looks as if the figures 1, 2 and 3 are overlapping. Please check if this is not a bug?

Author Response

The manuscript has been revised according to my comments and suggestions. However, there are still some errors in the manuscript:

Line 127, after (UAN) editorial error, two times parenthesis

We corrected this editing error.

Line 179, Line 335 under the tables there are a capital letter "P", elsewhere there is a lower case letter "p". Symbols needs to be unified.

We unified this symbol.

Figure 1B still has the lowercase letters. They shouldn't be there because there was no interaction S x F x T (this is confirmed in Table 1). In my pdf version it looks as if the figures 1, 2 and 3 are overlapping. Please check if this is not a bug?

The last version was without letter in Figure 1 and, as there was no interaction S x F x T (confirmed in Table 1, as you mentioned), we did not add symbol in Figure 1B.

Reviewer 5 Report

Dear authors, I appreciate the effort to improving the paper, but there are still some corrections to be made:
Figure 2 the letters of significance are still given but the interaction is not significant;
Figures 1 and 3 there are two superimposed images and in one continue to be letters of significance but the interaction is not significant;

Author Response

Dear authors, I appreciate the effort to improving the paper, but there are still some corrections to be made:
Figure 2 the letters of significance are still given but the interaction is not significant;
Figures 1 and 3 there are two superimposed images and in one continue to be letters of significance but the interaction is not significant.

The last version was without letter in Figure 2 and, as there was no interaction (confirmed in Table 1, as you mentioned). In the new version, you will find Figures 1 and 3 without superposed images.
